# Inactivation of *mrpigH* Gene in *Monascus ruber* M7 Results in Increased *Monascus* Pigments and Decreased Citrinin with *mrpyrG* Selection Marker

**DOI:** 10.3390/jof7121094

**Published:** 2021-12-19

**Authors:** Li Li, Na Xu, Fusheng Chen

**Affiliations:** 1Hubei International Scientific and Technological Cooperation Base of Traditional Fermented Foods, Huazhong Agricultural University, Wuhan 430070, China; lili1991@webmail.hzau.edu.cn (L.L.); xn@webmail.hzau.edu.cn (N.X.); 2College of Food Science and Technology, Huazhong Agricultural University, Wuhan 430070, China

**Keywords:** *Monascus ruber*, *mrpigH*, *Monascus* pigments, citrinin

## Abstract

*Monascus* pigments (MPs) have been used as food colorants for several centuries in Asian countries and are currently used around the world via Asian catering. The MPs biosynthetic pathway has been well-illustrated; however, the functions of a few genes including *mrpigH* in the MPs gene cluster of *M. ruber* M7 are still unclear. In the current study, *mrpigH* was disrupted in Δ*mrlig4*Δ*mrpyrG*, a highly efficient gene modification system, using *mrpyrG* as a selection marker, and Δ*mrpigH*Δ*mrlig4*Δ*mrpyrG*::*mrpyrG* and Δ*mrpigH*Δ*mrlig4*Δ*mrpyrG* have been obtained. Subsequently, their morphologies, biomasses, MPs and citrinin (CIT) production were analyzed, respectively. These results have revealed that the deletion of *mrpigH* has significant effects on the morphology and growth of *M. ruber* M7. Moreover, compared with *M. ruber* M7, the yields of MPs and CIT were drastically increased and decreased in *mrpigH* mutants, respectively.

## 1. Introduction

*Monascus* species are famous medicinal and edible filamentous fungi used in traditional fermentation in Asian countries, such as China, Japan, and the Korean Peninsula for nearly 2000 years [1,2]. At present, their fermented products, such as *Hongqu*, also called red fermented rice, red yeast rice and red mold rice, are widely used as food additives and nutraceutical supplements worldwide owing to their production of abundant beneficial secondary metabolites (SMs), such as *Monascus* pigments (MPs), monacolin K (MK) and γ-amino butyric acid (GABA) [1,3]. However, citrinin (CIT), a nephrotoxic mycotoxin produced by some strains of *Monascus* spp., restricts the application of *Monascus* fermented products [4].

*M. ruber* M7, which can produce MPs and CIT, without MK, was subjected to whole-genome sequencing analysis [5]. And the functions of most genes in the MPs gene cluster of *M. ruber* M7 have been investigated by gene manipulation [6]. However, there are a few genes in the MPs gene cluster of *M. ruber* M7, such as *mrpigH* and *mrpigI*, which have not been investigated [6,7]. In 2017, Balakrishnan et al. predicted that the *mppE* in *M*. *purpureus* KACC (highly homologous to *mrpigH* in *M. ruber* M7) encoded a reductase which can decrease orange pigments (OPs) and red pigments (RPs) in the biosynthesis of MPs [8]. In 2019, Chen et al. also guessed that MrPigH might contribute to reducing the carbon double bond of the precursor compounds to the typical yellow pigments (YPs) monascin and ankaflavin [7].

In this study, we firstly cloned *mrpigH* from *M. ruber* M7. Subsequently, *mrpigH* was disrupted in the highly efficient system Δ*mrlig4*Δ*mrpyrG* [9] using the *mrpyrG* selection marker, and two *mrpigH* deletion strains Δ*mrpigH*Δ*mrlig4*Δ*mrpyrG*::*mrpyrG* and Δ*mrpigH*Δ*mrlig4*Δ*mrpyrG* have been constructed. Finally, the morphologies, biomasses, MPs and CIT production of these *mrpigH* mutants were assessed. The results revealed that the deletion of *mrpigH* led to a dramatic reduction in biomass accumulation. Crucially, the inactivation of MrPigH resulted in an increase of MPs and a decrease of CIT.

## 2. Materials and Methods

### 2.1. Fungal Strains, Culture Media, and Growth Conditions

*M. ruber* M7 (CCAM 070120, Culture Collection of State Key Laboratory of Agricultural Microbiology, Wuhan, China), which can produce MPs and CIT, but no MK, is an original strain, which was isolated from *Hongqu* and preserved in our laboratory [10]. Δ*mrlig4*Δ*mrpyrG*, the highly efficient gene modification system for *M. ruber* M7, also named as the markerless disruption strain [9] was used to generate Δ*mrpigH*Δ*mrlig4*Δ*mrpyrG*::*mrpyrG* and Δ*mrpigH*Δ*mrlig4*Δ*mrpyrG*. All strains used in this study are described in Table 1. Strains were cultivated in PDA (potato dextrose agar) or minimal medium (MM, 2.0 g NH_4_Cl, 1 g (NH_4_)_2_SO_4_, 0.5 g KCl, 0.5 g NaCl, 1.0 g KH_2_PO_4_, 0.5 g MgSO_4_·7H_2_O, 0.02 g FeSO_4_·7H_2_O, 20.0 g glucose, distilled water to 1 L, pH 5.5). When required, 10 mM uridine and/or 0.75 mg/mL 5-fluoroorotic acid (5-FOA) were added. For observation of colonial and microscopic morphologies, four different types of media, PDA, malt extract agar (MA), Czapek yeast extract agar (CYA) and 25% glycerol nitrate agar (G25N) were utilized [11]. PDA was used for the analysis of MPs and CIT production. All strains were maintained on PDA slants at 28 °C.

### 2.2. Cloning and Analysis of mrpigH

A pair of primers, pigH F1–pigH R1 (Table 2), was designed to amplify *mrpigH* using Oligo 6 software (http://www.oligo.net/, accessed on 24 March 2021). PCR was carried out to amplify *mrpigH* from the genome of *M. ruber* M7. Amino acid sequence encoded by *mrpigH* was predicted using Softberry’s FGENESH program (http://www.softberry. com/, accessed on 24 March 2021), and the MrPigH functional regions were analyzed using the Pfam 33.1 program (http://pfam.xfam.org/, accessed on 24 March 2021). Homology of the deduced amino acid sequence was analyzed using the BLASTP program on the NCBI website (http://blast.ncbi.nlm.nih.gov/Blast.cgi, accessed on 24 March 2021).

### 2.3. Construction of Deletion Cassettes and Plasmids

The genomic DNA of *M. ruber* M7 used for PCR was isolated as described previously [12]. The mutant strains Δ*mrpigH*Δ*mrlig4*Δ*mrpyrG*::*mrpyrG* and Δ*mrpigH*Δ*mrlig4*Δ*mrpyrG* were constructed using site-directed homologous recombination. The *mrpigH* gene markerless deletion cassette (5′UTR-*mrpyrG*-5′-1UTR-3′UTR) was constructed by seamless cloning, and shown schematically in Figure 1a. The relative primer pairs were shown in Table 2.

The 5′ and 3′ flanking regions (993 bp and 775 bp, respectively) and the 5′-1 flanking region (531 bp) of *mrpigH* were amplified with the primers pigHpyrG 5F–pigHpyrG 5R, pigHpyrG 3F–pigHpyrG 3R and pigHpyrG 5F-1–pigHpyrG 5R-1, respectively. The 1.28-kb *mrpyrG* marker cassette was amplified from *M. ruber* M7 genomic DNA with the primer pair pigHpyrG pyrGeF–pigHpyrG pyrGeR. Then the four amplicons (5′ and 3′ regions, 5′-1 regions and *mrpyrG* expression fragment) were mixed at a 1:1:1:1 molar ratio and cloned into vector pBLUE-T using the seamless cloning and assembly kit (Beijing Zoman Biotechnology). Subsequently, both the cloned DNA fragment and the pCAMBIA3300 plasmid were digested with *Hind*III and *Xba*I, and ligated by T4 DNA ligase to generate plasmid pCPGPIGH for the *mrpigH* knock-out harbouring *mrpyrG* selection marker.

### 2.4. Deletion of mrpigH in Δmrlig4ΔmrpyrG Strain

The plasmid pCPGPIGH was transformed into *Agrobacterium tumefaciens* EHA105 using a freeze-thaw method [13]. Δ*mrlig4*Δ*mrpyrG*, a markerless disruption strain, was used as a host strain to delete *mrpigH* with the *mrpyrG* recyclable marker [9]. The *A. tumefaciens* clones containing pCPGPIGH were incubated for transformation with Δ*mrlig4*Δ*mrpyrG* to generate a *mrpigH* gene deletion mutant (Δ*mrpigH*Δ*mrlig4*Δ*mrpyrG*::*mrpyrG*) by minimal medium without uridine/uracil. The conidia of Δ*mrpigH*Δ*mrlig4*Δ*mrpyrG*::*mrpyrG* were collected and spread onto PDA with 0.75 mg/mL 5-FOA and 10 mM uridine. After incubated at 28 °C for 6 days, the surviving colonies were transferred to a new PDA under the same conditions for 4 days. The final surviving colonies were selected and verified by PCR. Selected transformants were designated as Δ*mrpigH*Δ*mrlig4*Δ*mrpyrG*.

### 2.5. MPs and CIT Analyses

*M. ruber* M7 can produce MPs and CIT, but no MK. Previous researches have shown that MPs mainly accumulate in the mycelia, while CIT exists in the media [14]. Therefore, the intracellular MPs and extracellular CIT were detected. 1 mL spores suspension (10^5^ cfu/mL) of each strain were inoculated on PDA plate coated with cellophane membranes and incubated at 28 °C for 11 days. 20 mg freeze-dried mycelia or media powder were suspended in 1 mL 80% (*v*/*v*) methanol solution, and subjected to 30 min ultrasonication treatment (KQ-250B, Kunshan, China). Then, the extraction solutions were separated by centrifugation at 10,000× *g* for 15 min and filtered with a 0.22 μm filter membrane for further analysis.

The pigments groups concentration was measured using a UV−vis UV-1700 spectrophotometer (Shimadzu, Tokyo, Japan) at 380, 470 and 520 nm which are the maximal absorption of yellow, orange, and red pigments, respectively. The results were expressed as optical density (OD) units per gram of dried mycelia multiplied by a dilution factor [15].

The CIT was detected on Waters ACQUITY UPLC BEH C18 column (2.1 mm × 100 mm, 1.7 µm, Waters) by fluorescence detector (Waters, Milford, MA, USA) in accordance with a previously described method [16].

### 2.6. Detection of the Relative Gene Expression Level in MPs and CIT Gene Clusters by RT-qPCR 

To analyze the influence of *mrpigH* deletion on gene expression in MPs and CIT gene cluster, Δ*mrpigH*Δ*mrlig4*Δ*mrpyrG*::*mrpyrG* and the wild-type strain (*M*. *ruber* M7) were selected for quantitative real-time PCR (RT-qPCR) detection. The Δ*mrpigH*Δ*mrlig4*Δ*mrpyrG* was lacked of *mrpyrG* and had to supply uridine, which might have had an effect on the yields of MPs and CIT.

One milliliter freshly harvested spores (10^5^ cfu/mL) of each strain were inoculated on PDA plate and incubated at 28 °C and samples were taken every other day from the third day to the ninth day. RT-qPCR was performed according to the method described by Liu et al. [13]. Beta-actin was used as a reference gene. The primers used in these analyses were listed in Appendix A.

## 3. Results

### 3.1. Sequence Analysis of mrpigH in M. ruber M7

A 1.24-kb fragment containing the putative *mrpigH* homolog was successfully amplified from the genomic DNA of *M.ruber* M7. Sequence prediction of *mrpigH* by Softberry’s FGENESH program has revealed that the putative *mrpigH* gene consists of a 1110 bp open reading frame (ORF) which consists of one exon and encodes 369-amino acids. A database search with Pfam 33.1 program has shown that MrPigH pertains to the alcohol dehydrogenase GroES-like domain. Besides, a database searched with NCBI-BLAST has been demonstrated that the deduced 369-amino acid sequences encoded by *mrpigH* share 65.31% similarity with the enoyl reductase (GenBank: PCH03974.1), 57.84% similarity with oxidoreductase of *Glonium stellatum* (GenBank: OCL03635.1), and 56.64% similarity with dehydrogenase of *Hyphodiscus hymeniophilus* (GenBank: KAG0646231.1). The specific function of *mrpigH* is still unclear.

### 3.2. Verification of the mrpigH Deletion in Δmrlig4ΔmrpyrG

The Δ*mrlig4*Δ*mrpyrG* strain is a promising host for efficient gene targeting in *M. ruber* M7 and analysis of biosynthesis of SMs. The deletion of *mrpigH* was executed in Δ*mrlig4*Δ*mrpyrG* with the *mrpyrG* marker (Figure 1a). After the plasmid pCPGPIGH harbouring *mrpyrG* was transformed into Δ*mrlig4*Δ*mrpyrG*, transformants without uridine/uracil auxotrophic were obtained and verified by PCR, and five of 28 transformants were *mrpigH*-deleted strains. As shown in Figure 1c, a 0.5-kb product was amplified when the genomic DNA of Δ*mrpigH*Δ*mrlig4*Δ*mrpyrG*::*mrpyrG* was used as template with primers pyrG F2-pyrG R2, while no DNA band was amplified using genome of the Δ*mrlig4*Δ*mrpyrG*. A 0.58-kb fragment of the *mrpigH* gene could be amplified from Δ*mrlig4*Δ*mrpyrG* using primers pigH F-pigH R, while no band was obtained from Δ*mrpigH*Δ*mrlig4*Δ*mrpyrG*::*mrpyrG*. Meanwhile, amplicons of Δ*mrpigH*Δ*mrlig4*Δ*mrpyrG*::*mrpyrG* (3.54 kb and 1.73 kb) and Δ*mrlig4*Δ*mrpyrG*(2.85 kb) differed in size when primers pigHpyrG 5F- pigHpyrG 3R annealing to homologous arms were used.

In the Δ*mrpigH*Δ*mrlig4*Δ*mrpyrG*::*mrpyrG* strain, the *mrpigH* deletion cassette contains two completely homologous sequences (5′UTR and 5′-1UTR) between the *mrpyrG* expression fragments. If Δ*mrpigH*Δ*mrlig4*Δ*mrpyrG*::*mrpyrG* was incubated on PDA plates containing 5-FOA, the strains that lost the *mrpyrG* expression fragments by homologous recombination should be selected. As predicted, Δ*mrpigH*Δ*mrlig4*Δ*mrpyrG* strains without the *mrpyrG* fragment could be isolated from Δ*mrpigH*Δ*mrlig4*Δ*mrpyrG*::*mrpyrG* after growth on PDA containing 0.75 mg/mL 5-FOA and 10 mM uridine. Total 12 putative Δ*mrpigH*Δ*mrlig4*Δ*mrpyrG* strains with 5-FOA resistance were obtained and analyzed, and one of them was shown as follows. In PCR analysis as shown in Figure 1e, a 0.58-kb fragment of the *mrpigH* gene and a 0.5-kb product of the *mrpyrG* gene could be amplified from *M. ruber* M7 using primers pigH F-pigH R and pyrG F2-pyrG R2, while nothing was obtained from Δ*mrpigH*Δ*mrlig4*Δ*mrpyrG*, respectively. Meanwhile, amplicons of Δ*mrpigH*Δ*mrlig4*Δ*mrpyrG* (1.73 kb) and *M. ruber* M7 (2.85 kb) differed in size when primers pigHpyrG 5F- pigHpyrG 3R annealing to homologous arms were used. Those results indicated that there was *mrpigH* markerless deletion was constructed in Δ*mrlig4*Δ*mrpyrG*.

### 3.3. Morphologies and Biomasses of mrpigH Mutants and M. ruber M7

To investigate whether the *mrpigH* was markerlessly deleted by *mrpyrG* in Δ*mrlig4*Δ*mrpyrG*, *M. ruber* M7 and its *mrpigH* markerless mutants were cultivated on PDA supplemented the appropriate additive (10 mM uridine for the uridine/uracil auxotrophy).The results (Figure 2a) revealed that Δ*mrpigH*Δ*mrlig4*Δ*mrpyrG* showed no growth on PDA, but it was able to grow on PDA with 0.75mg/mL 5-FOA and 10 mM uridine, while the growths of *M. ruber* M7 and Δ*mrpigH*Δ*mrlig4*Δ*mrpyrG*::*mrpyrG* were inhibited by the addition of 0.75 mg/mL 5-FOA to PDA, but could grow on PDA. Those results indicated that the *mrpigH* markerless mutants were successfully constructed.

To test the influence of deleting *mrpigH* on developmental processes, *M. ruber* M7 and Δ*mrpigH*Δ*mrlig4*Δ*mrpyrG*::*mrpyrG* were cultivated on different media (PDA, CYA, MA and G25N) to observe their colonial and microscopic characteristics. The results showed that the colonial morphologies of *mrpigH* mutants (Figure 2b) were obviously different from those of *M. ruber* M7 on different culture plates, especially on PDA plate, the Δ*mrpigH*Δ*mrlig4*Δ*mrpyrG*::*mrpyrG* showed slower growth rate and darker color. However, the microscopic morphologies, including conidia and cleistothecia, of Δ*mrpigH*Δ*mrlig4*Δ*mrpyrG*::*mrpyrG* was not dramatically different from those of *M. ruber* M7 on different culture plates (Figure 2c). Moreover, the biomasses of two *mrpigH* mutants apparently decreased compared with that of M7 on PDA in 5–11 d (Figure 2d).

### 3.4. Analysis of MPs and CIT Production

In order to evaluate the effect of detecting *mrpigH* on MPs and CITs during fermentation, the samples cultured for 3, 5, 7, 9 and 11 days on PDA were obtained and each test was repeated independently three times. OD values representing yellow, orange and red pigment production were determined using a spectrophotometer at 380 nm, 470 nm and 520 nm, respectively. As shown in Figure 3a–c, from the seventh day to the 11th day, *M. ruber* M7 produced much fewer MPs (including YPs, OPs and RPs) than 2 *mrpigH* mutants (Δ*mrpigH*Δ*mrlig4*Δ*mrpyrG*::*mrpyrG* and Δ*mrpigH*Δ*mrlig4*Δ*mrpyrG*), whereas the ability of producing MPs among 2 *mrpigH* mutants showed no obvious difference. After 11 days of cultivation, the YPs, OPs and RPs production in Δ*mrpigH*Δ*mrlig4*Δ*mrpyrG*::*mrpyrG* and Δ*mrpigH*Δ*mrlig4*Δ*mrpyrG* were 2.04–2.22, 8.76–10.06, 4.29–4.69 times those of *M. ruber* M7, respectively.

As to CIT, the UPLC has been performed to detect the production during fermentation. As shown in Figure 3d, CIT produced by *M. ruber* M7 and all *mrpigH* mutants showed an obvious difference. At the end of the 11 days of fermentation, CIT production in Δ*mrpigH*Δ*mrlig4*Δ*mrpyrG*::*mrpyrG* and Δ*mrpigH*Δ*mrlig4*Δ*mrpyrG* decreased dramatically, and was two to three orders of magnitude less than that of *M.ruber* M7. Among 2 *mrpigH* mutants, CIT production in Δ*mrpigH*Δ*mrlig4*Δ*mrpyrG* was higher than that of Δ*mrpigH*Δ*mrlig4*Δ*mrpyrG*::*mrpyrG*. The possible cause was that the Δ*mrpigH*Δ*mrlig4*Δ*mrpyrG* lacked *mrpyrG* and had to supply uridine, which might have an effect on the yields of CIT. 

### 3.5. The Genes’ Expression in MPs and CIT Gene Clusters from mrpigH Mutants

The gene expression in MPs and CIT gene clusters in Δ*mrpigH*Δ*mrlig4*Δ*mrpyrG*::*mrpyrG* and *M. ruber* M7, were analyzed by RT-qPCR. As shown in Figure 4, the relative expression levels of *mrpigA*, *mrpigB*, *mrpigC*, *mrpigD*, *mrpigE*, *mrpigF*, *mrpigG*, *mrpigJ*, *mrpigK*, *mrpigM*, *mrpigN*, *mrpigO* and *mrpigP* in Δ*mrpigH*Δ*mrlig4*Δ*mrpyrG*::*mrpyrG* were obviously higher than that of *M. ruber* M7 on the fifth to ninth days. Therefore, the deletion of *mrpigH* increased the majority of MPs gene expression level, which might correspond with the enhanced MPs production.

In a previous study, Li et al. [17] considered the CIT biosynthetic gene cluster in *M*. *aurantiacus* including 16 genes. He and Cox [18] demonstrated that a minimal set of conserved genes were involving in CIT biosynthesis, including: oxydoreductase (*mrl7*), dehydrogenase (*mrl6*), glyoxylase-like domain (*mrl5*), NAD(P)+ dependent aldehyde dehydrogenase (*mrl4*), transcriptional regulator (*mrl3*), Fe(II)-dependent oxygenase (*mrl2*), serine hydrolase (*mrl1*), non-reducing polyketide synthase (*mrpks*),and major facilitator superfamily (MFS) protein (*mrr1*). As shown in Figure 5, the relative expression levels of *mrl6*, *mrl5*, *mrl4*, *mrl2*, *mrl1*, *mrpks* and *mrr1* were dramatically lower than those of *M. ruber* M7 at 3rd and 5th day. Therefore, the deletion of *mrpigH* decreased the expression level of core CIT genes, which might correspond with the reduced CIT production. 

## 4. Discussion

*Monascus* spp. have been widely used in food fermentation for nearly 20 centuries in East and Southeast Asian countries due to their ability of producing vivid MPs [11,19], which are a complex mixture of secondary metabolites (SMs) with a tricyclic azaphilone scaffold, produced via the polyketide pathway by a few filamentous fungi such as *Monascus* spp. and *Penicillium* spp. [3,20,21]. MPs are biosynthesized by their gene cluster, which contains 16 genes (*mrpigA-mrpigP*) in *M.ruber* M7 [6,7].

In the current study, in order to explore the function of *mrpigH**,* we used the *Δmrlig4*Δ*mrpyrG* as the starting strain to generate two kinds of *mrpigH* disruptants, namely Δ*mrpigH*Δ*mrlig4*Δ*mrpyrG*::*mrpyrG* and Δ*mrpigH*Δ*mrlig4*Δ*mrpyrG*. The colonial and microbiological phenotypes and biomasses of the *mrpigH* mutants showed an obviously difference from those of *M. ruber* M7 (Figure 2b, d). In particular, the biomasses of *mrpigH* mutants accumulated more slowly than that of *M ruber* M7. Our laboratory previously constructed MPs gene knockout strains with the resistance selection markers (*hph*/*neo*) in M7, and found that the morphologies and biomasses of *mrpigC*, *mrpigE*, *mrpigF*, *mrpigM* and *mrpig*O knockouts were comparable to those of *M. ruber* M7, whereas the growth rates of *mrpigA*, *mrpigJ*, and *mrpigK* knockouts were increased, and the deletion of *mrpigN* resulted in a reduction in biomass accumulation compared with *M. ruber* M7 [22].

Moreover, we have discovered that the deletion of *mrpigH* in *M. ruber* M7 can enhance YPs, OPs and RPs (Figure 3a–c), which is mostly consistent with the results obtained in *M. purpureus* [8]. We also found that deletion of *mrpigH* dramatically decreased the CIT production (Figure 3d), which is a promising scheme for control of CIT. CIT is a kind of mycotoxin produced via the polyketide pathway by filamentous fungi, mainly by *Monascus* spp. and *Penicillium* spp. [23,24,25]. To this end, we investigated the changes in the expression levels of MPs and CIT genes in their gene clusters of *mrpigH* mutants, and found that the relative gene expression levels almost corresponded with the increase of MPs and decrease of CIT (Figure 4 and Figure 5).

The deep study about the reason for increasing the pigment content while reducing the CIT production by knocking out *mrpigH* need to be investigated further. In early research, both MPs and CIT were considered to be derived from polyketide pathways [26]. Lately, two hypotheses about the biosynthetic pathways of MPs and CIT were put forward. One of them is based on metabolic pathways; the MPs and CIT shared a common pathway to a certain branch [26]. The other is based on the analysis of the whole genome sequence; their biosynthetic gene clusters had been found separately, thereby, they belonged to two different pathways [6,18]. 

Previous results suggested that the yields of CIT and MPs in *Monascus* spp. could affect each other. For example, Xie et al. (2013) and Liu et al. (2014) separately identified that the overexpression of *mrpigB* and *mrpigE* in MPs gene cluster of *M*. *ruber* M7 resulted in a reduction of CIT production [13,27]. Meanwhile, Liang et al. (2017) obtained a mutant of a putative glyoxalase (*orf6*) in CIT gene cluster of *M*. *purpureus**,* and found that the deletion of *orf6* could improve the MPs and CIT yields at the same time [28]. A recent study on the biosynthesis of MPs found that acetyl-CoA and malonyl-CoA are catalyzed by a sequence of enzymes to produce MPs precursors. Then the pathway bifurcates into two branches. First, the MPs precursors were reduced by a reductase (MrPigH and/or GME3457, which is encoded outside of the MPs gene cluster in *M. ruber* M7), yielding the typical YPs monascin and ankaflavin. The other branch of the pathway produces the typical OPs rubropunctatin and monascorubrin by a FAD-dependent oxidoreductase (MrPigF in *M. ruber* M7). Furthermore, rubropunctatin and monascorubrin were converted into RPs rubropunctamine and monascorubramine through an amination reaction [6,7]. Due to the deletion of *mrpigH*, the relative expression of *mrpigF* dramatically increased (Figure 4), which was beneficial for the OPs and RPs production, and the increase in relative yields of OPs and RPs was greater than that of YPs. Recently, Li et al. (2020) reported that MPs biosynthetic gene cluster was a composite supercluster, and the naphthoquinone (monasone) gene cluster was embedded in the MPs gene cluster, and speculated that MrPigH was essential in the biosynthesis of naphthoquinone, but it was a supplemental enzyme in the biosynthesis of MPs [29]. As a result, the *mrpigH* knockout’s blocking of the naphthoquinone biosynthesis pathway might result in naphthoquinone reduction, and more substrates and intermediates were utilized to synthesize MPs, including YPs. In terms of CIT production, He Yi et al. [18] revealed a minimal set of conserved genes involved in CIT biosynthesis, which included nine genes (*mrl7*, *mrl6*, *mrl5*, *mrl4*, *mrl3*, *mrl2*, *mrl1*, *mrpks* and *mrr1*). When *mrpigH* was deleted, the majority of the genes in the cluster (*mrl6*, *mrl5*, *mrl4*, *mrl2*, *mrl1*, *mrpks* and *mrr1*) were obviously down-regulated (Figure 5), which could explain why CIT production has decreased in this study.

In conclusion, the *mrpigH* gene pertains to the MPs biosynthetic gene cluster of *M. ruber* M7 and plays a remarkable role in the biosynthesis of MPs and CIT. The disruption of *mrpigH* had very little effect on the microscopic morphologies, while the *mrpigH* mutants showed slower biomass accumulation and darker color on PDA. Compared with *M*. *ruber* M7, the YPs, OPs and RPs production in the *mrpigH* mutants (Δ*mrpigH*Δ*mrlig4*Δ*mrpyrG*::*mrpyrG* and Δ*mrpigH*Δ*mrlig4*Δ*mrpyrG*) increased dramatically. However, the CIT production of the *mrpigH* mutants decreased drastically. This work will make some contribution to the regulation of MPs and CIT production in *M. ruber* M7.

## Figures and Tables

**Figure 1 jof-07-01094-f001:**
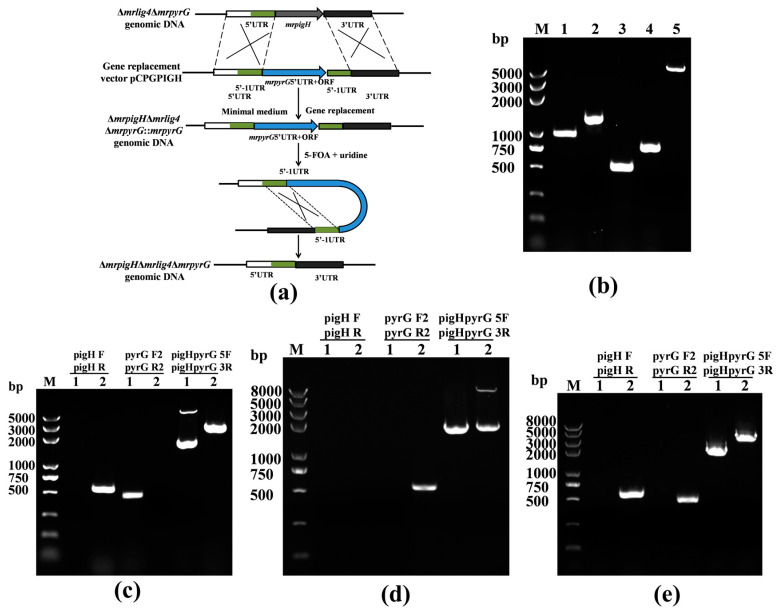
Markerless deletion of *mrpigH* in Δ*mrlig4*Δ*mrpyrG*. (**a**) Schematic representation of the homologous recombination strategy yielding *mrpigH* markerless deletion strains. (**b**) Construction of *mrpigH* disruption construct by Seamless Cloning and assembly method. Lane 1, 5′ flanking region of *mrpigH*; lane 2, 5′ flanking region of *mrpyrG* plus *mrpyrG* ORF regions; lane 3, 5′-1 flanking region of *mrpigH*; lane 4, 3′ flanking region of *mrpigH*; lane 5, deletion cassette product. (**c**) Confirmation of *mrpigH* homologous recombination events. Three primer pairs were used and PCR amplifications showed distinct bands in different strains. Lane 1, the Δ*mrpigH*Δ*mrlig4*Δ*mrpyrG*::*mrpyrG* strain; lane 2, the Δ*mrlig4*Δ*mrpyrG* strain. (**d**) Confirmation of *mrpyrG* homologous recombination events in Δ*mrpigH*Δ*mrlig4*Δ*mrpyrG*::*mrpyrG* strain. Lane 1, the Δ*mrpigH*Δ*mrlig4*Δ*mrpyrG* strain; lane 2, the Δ*mrpigH*Δ*mrlig4*Δ*mrpyrG*::*mrpyrG* strain. (**e**) Confirmation of *mrpigH* markerless deletion in Δ*mrpigH*Δ*mrlig4*Δ*mrpyrG* strain. Lane 1, the Δ*mrpigH*Δ*mrlig4*Δ*mrpyrG* strain; lane 2, the wild-type strain.

**Figure 2 jof-07-01094-f002:**
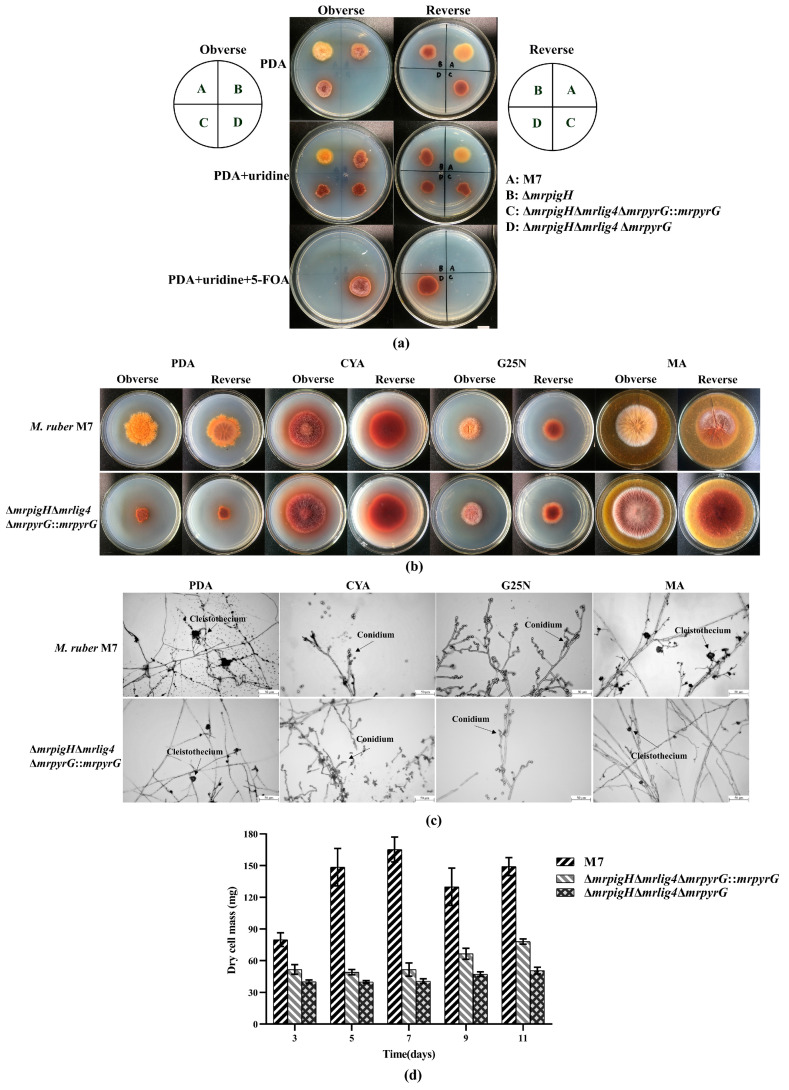
Morphologies and biomasses of *mrpigH* mutants and *M. ruber* M7. (**a**) Growth of *M. ruber* M7 and *mrpigH* mutants on PDA, PDA with 10 mM uridine, PDA with 10 mM uridine and 0.75 mg/mL 5-FOA for 5 days at 28 °C. A, *M. ruber* M7; B, Δ*mrpigH*, *mrpigH* was disrupted in *M. ruber* M7 with the *hph* selection marker [9]; C, Δ*mrpigH*Δ*mrlig4*Δ*mrpyrG*::*mrpyrG*; D, Δ*mrpigH*Δ*mrlig4*Δ*mrpyrG*; (**b**) Colonial morphologies of *M. ruber* M7 and Δ*mrpigH*Δ*mrlig4*Δ*mrpyrG*::*mrpyrG* on PDA, CYA, MA and G25N plates for 10 days at 28 °C; (**c**) Cleistothecia and conidia formation of *M. ruber* M7 and Δ*mrpigH*Δ*mrlig4*Δ*mrpyrG*::*mrpyrG* on different plates (PDA, CYA, G25N, and MA) for 7 days at 28 °C; (**d**) Biomass (dry cell weight) of *M. ruber* M7 and *mrpigH* mutants on PDA plates at 28 °C. The *error bar* represents the standard deviation between the three repeats.

**Figure 3 jof-07-01094-f003:**
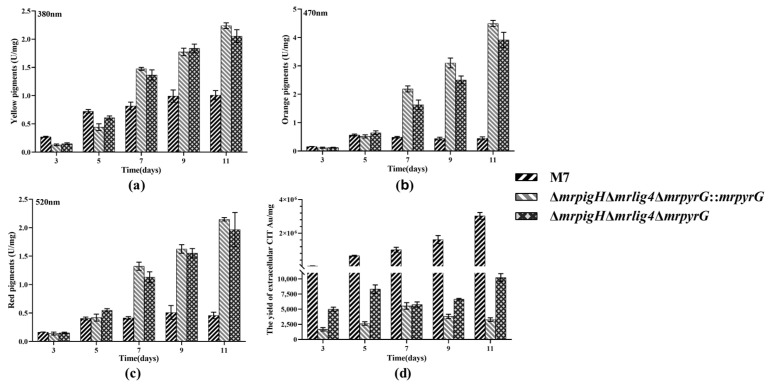
Production of intracellular MPs and extracellular CIT by *M. ruber* M7 and *mrpigH* mutants on the PDA supplied with/without uridine. (**a**) The yield of yellow pigments; (**b**) The yield of orange pigments; (**c**) The yield of red pigments; (**d**) The yield of CIT. The *error bar* represents the standard deviation between the three repeats.

**Figure 4 jof-07-01094-f004:**
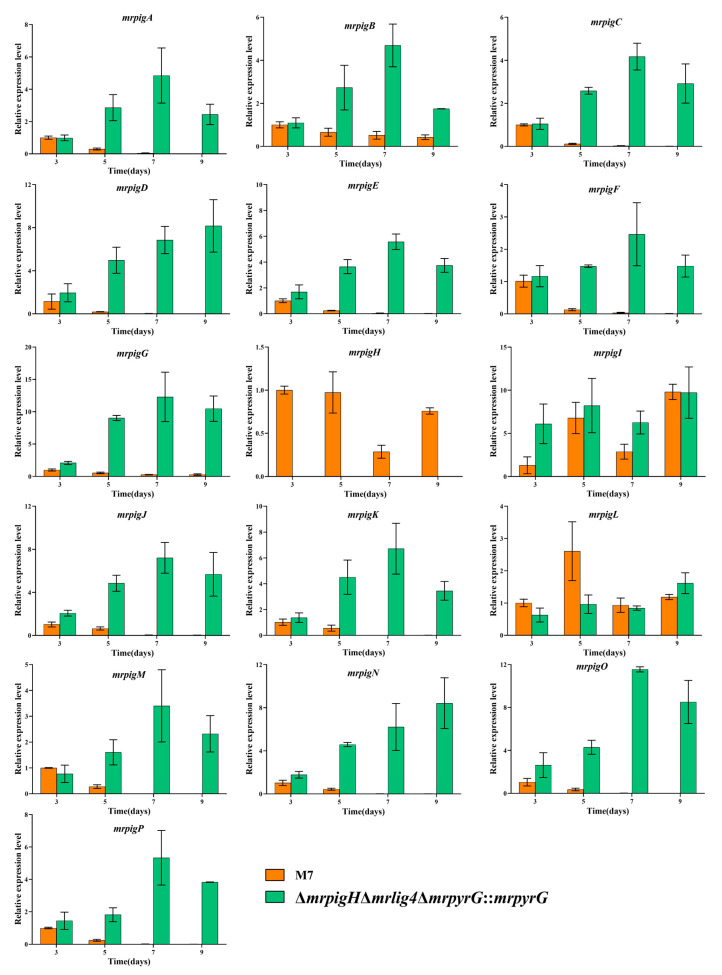
RT-qPCR analysis of the *mrpigA-mrpigP* in M7 and Δ*mrpigH*Δ*mrlig4*Δ*mrpyrG*::*mrpyrG*. *mrpigA*: Non-reducing polyketide synthase; *mrpigB*: Transcription factor; *mrpigC*: C-11-Ketoreductase; *mrpigD*: 4-O-Acyltransferase; *mrpigE*: NAD(P)H-dependent oxidoreductase; *mrpigF*: FAD-dependent oxidoreductase; *mrpigG*: Serine hydrolase; *mrpigH*: Enoyl reductase; *mrpigI*: Transcription factor; *mrpigJ*: FAS subunit alpha; *mrpigK*: FAS subunit beta; *mrpigL*: Ankyrin repeat protein; *mrpigM*: *O*-Acetyltransferase; *mrpigN*: FAD-dependent monooxygenase; *mrpigO*: Deacetylase; *mrpigP*: MFS multidrug transporter; M7 and Δ*mrpigH*Δ*mrlig4*Δ*mrpyrG*::*mrpyrG* were incubated in PDA on the 3rd, 5th, 7th, 9th day at 28 °C The beta-actin gene was used as control. The error bars indicate the standard deviations of three independent cultures.

**Figure 5 jof-07-01094-f005:**
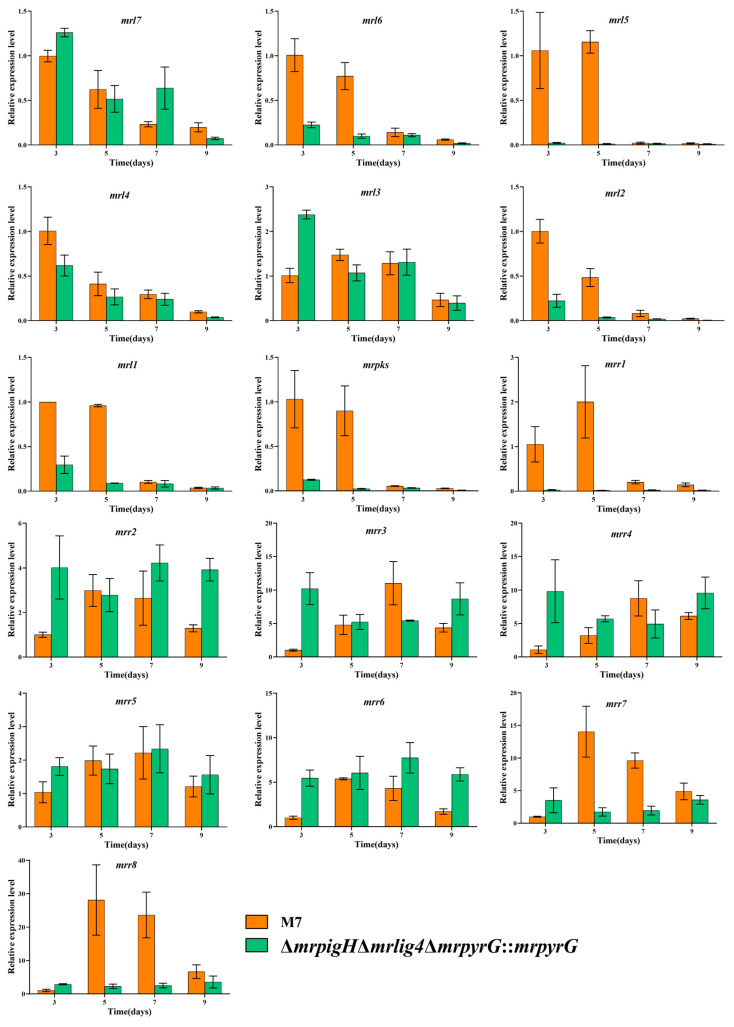
RT-qPCR analysis of the *mrl7-mrr8* in M7 and Δ*mrpigH*Δ*mrlig4*Δ*mrpyrG*::*mrpyrG*. *mrl7*: Oxydoreductase; *mrl6*: Dehydrogenase; *mrl5*: Glyoxylase-like domain; *mrl4*: NAD(P)+ dependent aldehyde dehydrogenase; *mrl3*: Transcriptional regulator; *mrl2*: Fe(II)-dependent oxygenase; *mrl1*: Serine hydrolase; *mrpks*: Non-reducing PKS; *mrr1*: Major facilitator superfamily (MFS) protein; *mrr2*: Histidine phosphatase; *mrr3*: Unknown protein; *mrr4*: WD40 protein; *mrr5*: Carbonic anhydrase; *mrr6*: Unknown protein; *mrr7*: Enoyl-(acyl carrier protein) reductase; *mrr8*: AMP-binding enzyme; M7 and Δ*mrpigH*Δ*mrlig4*Δ*mrpyrG*::*mrpyrG* were incubated in PDA on the 3rd, 5th, 7th, 9th day at 28 °C. The beta-actin gene was used as control. The error bars indicate the standard deviations of three independent cultures.

**Table 1 jof-07-01094-t001:** *M. ruber* strains constructed and used in this study.

Strain	Parent	Source
*M. ruber* M7	M7 [10]	Red fermented rice
Δ*mrpigH*Δ*mrlig4* Δ*mrpyrG*::*mrpyrG*	Δ*mrlig4*Δ*mrpyrG* [9]	This study
Δ*mrpigH*Δ*mrlig4* *mrpyrG*	Δ*mrpigH*Δ*mrlig4* Δ*mrpyrG*::*mrpyrG*	This study

**Table 2 jof-07-01094-t002:** Primers used in this study.

Names	Sequences (5′→3′)	Descriptions
pigHpyrG 5F	GATATCGAATTCCCAATACT CGTTACCCCGTCCAAGATGG	For amplification of the 993 bp of 5′ flanking regions of the *mrpigH*
pigHpyrG 5R	CGGTGGCAGTCGAAGGGGCA	
pigH pyrG pyrGeF	TGCCCCTTCGACTGCCACCG GATTATCGTATAGAGCAATA	For the expression of the1276 bp of the *mrpyrG*
pigH pyrG pyrGeR	TCACTGGTTCTTACAGCCGT	
pigHpyrG 5-1F	ACGGCTGTAAGAACCAGTGA CGCACACACGTTTCGCACGG	For amplification of the 531 bp of 5′-1 flanking regions of the *mrpigH*
pigHpyrG 5-1R	CGGTGGCAGTCGAAGGGGCA	
pigHpyrG 3F	TGCCCCTTCGACTGCCACCG GGCTGGATGCTGCATGTTTT	For amplification of the 775 bp of 3′ flanking regions of the *mrpigH*
pigHpyrG 3R	CTGCAGGAATTCCCAATACT CGCCGAAGCCCCCTTCCTCT	
pigH F	GTGCTGGTGCCCGACCTGAC	For amplification of the 583 bp of the partial *mrpigH*
pigH R	CGAAGATGAAATTCGACTTGA	
pyrG F2	GTGCATACTCTACAGAT	For amplification of the 498 bp of the partial *mrpyrG*
pyrG R2	CCAAGAAGACGAATGTGA	

Labeled with dotted lines letters are nucleotide sequences of pBLUE-T vector; labeled with single underline letters are nucleotide sequences of 5′UTR of *mrpigH*; labeled with double underline letters are nucleotide sequences of *mrpyrG* gene; labeled with wavy line letters are nucleotide sequences of 5′-1UTR of *mrpigH*.

## Data Availability

The raw data supporting the conclusions of this manuscript will be made available by the authors, without undue reservation, to any qualified researcher.

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
