# Peer review of "Inactivation of mrpigH Gene in Monascus ruber M7 Results in Increased Monascus Pigments and Decreased Citrinin with mrpyrG Selection Marker"

_jof, 2021, doi:10.3390/jof7121094_

Round 1

Reviewer 1 Report

  1. The title can be reframed as: “Inactivation of mrpigH gene in Monascus ruber M7 results in increased pigment but decreased citrinin production”
  2. In title why gene name is capitalized?
  3. The author’s have used the term “Significant” multiple times (Line No. 219, 239, and so on) but for most of the cases, they did not provide any fold change value or statistical analysis. 
  4. The author’s mentioned the slow growth of mrpigH mutants in comparison to the parent ruber M7. But both the strains produce colonies of almost equal size in different media except PDA. If slow growth results from the deletion of mrpigH then why it is not manifested in all media. Please discuss.
  5. 2 d shows an abrupt change in dry cell mass on day 9 for M. ruber M7 and also the mutant. Please justify.
  6. The authors did not provide any HPLC-generated graph for citrinin or pigment analysis. For better presentation and authenticity of the data, it can be provided.
  7. 3 d: Authors used the unit AU/mg for Citrinin. It will be more convenient if they use ppm or µg/mg.
  8. In the case of ΔmrpigHΔmrlig4ΔmrpyrG::mrpyrG, the result showsdecrease in citrinin content in the media after day 7 ( fig. 3d). What about the previously accumulated citrinin? Is it degraded or somehow utilized by the strain? Please justify.
  9. Authors should provide a correlation study for gene expression and phenotypic expression for both pigment and citrinin.

Reviewer 2 Report

Ongoing work over several years has been revealing the biosynthetic steps to the monascus pigments that are important compounds in food preparation. These pigments are made by various fungi, but classically Monascus ruber and closely related organisms are the best studied. The M. ruber pathway is already well elucidated, but the roles of some proteins encoded by genes in the pig BGC are still mysterious. Here KOs of mrpigH are made. Overall this is a very straight-forward story. The gene is knocked out using classical methods and this is proven genetically. The strain is then investigated morphologically (fine) and chemically. The gene encodes an enoyl-reductase-like protein and has previously been proposed as involved in later steps of the pathway (e.g. ref 6). Previous work (e.g. ref 6) has mapped out the pathway in some detail and also provided significant analytical chemistry (e.g LCMS) data on the intermediates.

In this MS the chemical analysis is much poorer - no HPLC/LCMS analysis is given of changes to the pig pathway, only some broad uv analysis of yellow, orange and red pigments with no structural analysis - although analysis of competing citrinin  was done by HPLC. This means that the chemical role of PigH cannot be assessed. Given the high quality of previous work by this group, this is a great weakness.

The titre of citrinin was also greatly reduced. While a convincing molecular mechanism was not suggested to explain this, it is an interesting observation. However to suggest this might be a good way to reduce citrinin titres is disingenuopus - the way to reduce citrinin titres is to KO the citrinin PKS....

Overall I found this paper somewhat disappointing given the high quality of previous work in this area. Lack of proper LCMS data means that the actual function of pigH remains unknown, and it would have been simple to extend the previous very good analytical work.

minor points

  1. line 43 - should be mrpigH and not mpigH;

2. line 45 and line 317 - "two" not "2";

3. Figure 2 - please keep legend with figure

Author Response

Response to Reviewer 2 Comments

Dear Reviewer

Thanks for your letter and comments on the manuscript entitled “Inactivation of MrPigH in Monascus ruber M7 results in its Monascus pigments increase and citrinin decrease with mrpyrG selection marker” (Manuscript ID: jof-1491459). Those comments are all valuable and very helpful for revising and improving this paper. We have studied comments carefully and have made correction which we hope meet with approval.

Thank you for your comment. About the ref 6” Orange, red, yellow: biosynthesis of azaphilone pigments in Monascus fungi. Chem Sci 2017, 8, 4917-4925”, the majority of MPs genes have been analyzed in this article, and the MPs biosynthetic pathway has been presented. However, the functions of some genes are still unknown, including mrpigH. The mrpigH mutants was not obtained and the function of mrpigH need to be investigated.

In our article, we successfully obtained the mrpigH deletion strain and investigated its characteristics, concluding that the deletion of mrpigH would result in an increase in pigment groups (color value) and a decrease in CIT. Owing to the limitations of the experimental conditions, no chromatographic data about MPs was collected. However, the available pigment groups data (optical density, OD) and CIT data can be used to deduce the experimental results. As your comment, more research into chromatographic data about MPs will assist to clarify the involvement of mrpigH in the MPs biosynthesis.

In terms of reducing the production of CIT, it is common assumed that knocking out genes in the CIT biosynthetic gene cluster will result in a considerable reduction in CIT production, and the findings are as predicted. However, the MPs production is similarly diminished once the CIT gene is knocked out [1], especially for those commercial MPs fermentation strains (data was not shown). These results aren’t what we're looking for. In the actual manufacturing of MPs, we intend to ensure the MPs production and minimize the CIT production. In our article, we discovered that the deletion of mrpigH can resulted in increased MPs and decreased CIT. This finding is meaningful for MPs manufacturing. At the same time, the relationship between the MPs and CIT biosynthetic pathways is worth investigating.

As your comment, the discussion section of the article was expanded to include information on how mrpigH regulates MPs and citrinin.

“A recent study on the biosynthesis of MPs found that acetyl-CoA and malonyl-CoA are catalyzed by a sequence of enzymes to produce MPs precursors. Then the pathway bifurcates into two branches. First, the MPs precursors were reduced by a reductase (MrPigH and/or GME3457, which is encoded outside of the MPs gene cluster in M. ruber M7), yielding the typical YPs monascin and ankafiavin. The other branch of the pathway produces the typical OPs rubropunctatin and monascorubrin by a FAD-dependent oxidoreductase (MrPigF in M. ruber M7). Furthermore, rubropunctatin and monascorubrin were converted into RPs rubropunctamine and monascorubramine through an amination reaction [2,3]. Due to the deletion of mrpigH, the relative expression of mrpigF dramatically increased (Figure 4), which was beneficial for the OPs and RPs production, and the increase in relative yields of OPs and RPs was greater than that of YPs. Recently, Li et al. (2020) reported that MPs biosynthetic gene cluster was a composite supercluster, and the naphthoquinone (monasone) gene cluster was embedded in the MPs gene cluster, and speculated that MrPigH was essential in the biosynthesis of naphthoquinone, but it was a supplemental enzyme in the biosynthesis of MPs [4]. As a result, the mrpigH knockout's blocking of the naphthoquinone biosynthesis pathway might result in naphthoquinone reduction, and more substrates and intermediates were utilized to synthesize MPs, including YPs. In terms of CIT production, He Yi et al. [5] revealed a minimal set of conserved genes involved in CIT biosynthesis, which included nine genes (mrl7, mrl6, mrl5, mrl4, mrl3, mrl2, mrl1, mrpks and mrr1). when mrpigH was deleted, the majority of the genes in the cluster (mrl6, mrl5, mrl4, mrl2, mrl1, mrpks and mrr1) were obviously down-regulated (Figure 5), which could explain why CIT production has decreased in this study.”

Reference

  1. Li, Y.P.; Tang, X.; Wu, W.; Xu, Y.; Huang, Z.B.; He, Q.H. The ctnG gene encodes carbonic anhydrase involved in mycotoxin citrinin biosynthesis from Monascus aurantiacus. Food Addit Contam A 2015, 32, 577-583.
  2. Chen, W.P.; Feng, Y.L.; Molnar, I.; Chen, F.S. Nature and nurture: confluence of pathway determinism with metabolic and chemical serendipity diversifies Monascus azaphilone pigments. Nat Prod Rep 2019, 36, 561-572.
  3. Chen, W.P.; Chen, R.; Liu, Q.P.; He, Y.; He, K.; Ding, X.L.; Kang, L.J.; Guo, X.X.; Xie, N.N.; Zhou, Y.X., et al. Orange, red, yellow: biosynthesis of azaphilone pigments in Monascus fungi. Chem Sci 2017, 8, 4917-4925.
  4. Li, M.; Kang, L.J.; Ding, X.L.; Liu, J.; Liu, Q.P.; Shao, Y.C.; Molnar, I.; Chen, F.S. Monasone Naphthoquinone Biosynthesis and Resistance in Monascus Fungi. Mbio 2020, 11.
  5. He, Y.; Cox, R.J. The molecular steps of citrinin biosynthesis in fungi. Chem Sci 2016, 7, 2119-2127.

Minor point

 Point 1: line 43 - should be mrpigH and not mpigH

 Response 1:  very thank reviewer for careful reading, the mpigH was changed to mrpigH.

Point 2: line 45 and line 317 - "two" not "2";

Response 2: thank you for your comment. the number 2 on line 45 and line 317 has been replaced with two.

Point 3: Figure 2 - please keep legend with figure

 Response 3: Thank you for your careful reading. The original legend of figure 2 is “Figure 2. Morphologies and biomasses of mrpigH mutants and M. ruber M7. (a) Growth of M. ruber M7 and mrpigH mutants on PDA, PDA with 10 mM uridine, PDA with 10 mM uridine and 0.75 mg/mL 5-FOA for 5 days at 28 °C. A, M. ruber M7; B, ΔmrpigH, mrpigH was disrupted in M. ruber M7 with the hph selection marker [9]; C, ΔmrpigHΔmrlig4ΔmrpyrG::mrpyrG; D, ΔmrpigHΔmrlig4ΔmrpyrG; (b) Colonial morphologies (at 10 days) on PDA, CYA, MA and G25N plates at 28 °C; (c) Cleistothecia and conidia formation (at 7 days) of  M. ruber M7 and mrpigH mutants on different plates (PDA, CYA, G25N, and MA) at 28 °C; (d) Biomass (dry cell weight). The error bar represents the standard deviation between the three repeats.”

The original legend has been replaced by “Figure 2. Morphologies and biomasses of mrpigH mutants and M. ruber M7. (a) Growth of M. ruber M7 and mrpigH mutants on PDA, PDA with 10 mM uridine, PDA with 10 mM uridine and 0.75 mg/mL 5-FOA for 5 days at 28 °C. A, M. ruber M7; B, ΔmrpigH, mrpigH was disrupted in M. ruber M7 with the hph selection marker [9]; C, ΔmrpigHΔmrlig4ΔmrpyrG::mrpyrG; D, ΔmrpigHΔmrlig4ΔmrpyrG; (b) Colonial morphologies of  M. ruber M7 and ΔmrpigHΔmrlig4ΔmrpyrG::mrpyrG on PDA, CYA, MA and G25N plates for 10 days at 28 °C; (c) Cleistothecia and conidia formation of M. ruber M7 and ΔmrpigHΔmrlig4ΔmrpyrG::mrpyrG on different plates (PDA, CYA, G25N, and MA) for 7 days at 28 °C; (d) Biomass (dry cell weight) of M. ruber M7 and mrpigH mutants on PDA plates at 28 °C. The error bar represents the standard deviation between the three repeats.”

Round 2

Reviewer 2 Report

The authors have addressed all the major points. I still feel a major weakness of the present work is the lack of proper analytical chemistry, but perhaps the authors will publish this elsewhere. A couple of minor points:

  1. It would be very helpful to move text so that the legend for figure 2 appears on the same page as figure 2.

2. Legend to Fig 4 refers to a polyketone synthase - this should be "polyketide synthase"

3. Ankaflavin is incorrectly spelled as ankafiavin at line 331; line 346 the first word of the sentence "when" should be capitalised

Author Response

Response to Reviewer 2 Comments

Dear Reviewer

Thanks for your letter and comments on the manuscript entitled “Inactivation of MrPigH in Monascus ruber M7 results in its Monascus pigments increase and citrinin decrease with mrpyrG selection marker” (Manuscript ID: jof-1491459). Those comments are all valuable and very helpful for revising and improving this paper. We have studied comments carefully and have made correction which we hope meet with approval.

Point 1: It would be very helpful to move text so that the legend for figure 2 appears on the same page as figure 2

Response 1:  very thank reviewer for your comment. I have adjusted the position of the legend in Figure 2 so that it can be displayed on the same page as the picture in Figure 2

Point 2: Legend to Fig 4 refers to a polyketone synthase - this should be "polyketide synthase"

Response 2:  thank you for your careful reading. The word “polyketone” has been replaced with “polyketide”.

Point 3: Ankaflavin is incorrectly spelled as ankafiavin at line 331; line 346 the first word of the sentence "when" should be capitalised

Response 3:  thank you for your careful reading. The word “ankafiavin” at line 331 has been changed to “ankaflavin”. The first word of the sentence “when” at line 346 has been capitalized.